# Modelling and Molecular Dynamics Predict the Structure and Interactions of the Glycine Receptor Intracellular Domain

**DOI:** 10.3390/biom13121757

**Published:** 2023-12-07

**Authors:** James R. E. Thompson, Christopher A. Beaudoin, Sarah C. R. Lummis

**Affiliations:** Department of Biochemistry, University of Cambridge, Cambridge CB2 1QW, UK

**Keywords:** pentameric ligand-gated ion channel Cys-loop receptor, gephyrin, binding site

## Abstract

Glycine receptors (GlyRs) are glycine-gated inhibitory pentameric ligand-gated ion channels composed of α or α + β subunits. A number of structures of these proteins have been reported, but to date, these have only revealed details of the extracellular and transmembrane domains, with the intracellular domain (ICD) remaining uncharacterised due to its high flexibility. The ICD is a region that can modulate function in addition to being critical for receptor localisation and clustering via proteins such as gephyrin. Here, we use modelling and molecular dynamics (MD) to reveal details of the ICDs of both homomeric and heteromeric GlyR. At their N and C ends, both the α and β subunit ICDs have short helices, which are major sites of stabilising interactions; there is a large flexible loop between them capable of forming transient secondary structures. The α subunit can affect the β subunit ICD structure, which is more flexible in a 4α_2_:1β than in a 4α_1_:1β GlyR. We also explore the effects of gephyrin binding by creating GlyR models bound to the gephyrin E domain; MD simulations suggest these are more stable than the unbound forms, and again there are α subunit-dependent differences, despite the fact the gephyrin binds to the β subunit. The bound models also suggest that gephyrin causes compaction of the ICD. Overall, the data expand our knowledge of this important receptor protein and in particular clarify features of the underexplored ICD.

## 1. Introduction

Glycine receptors (GlyRs) are members of the Cys-loop superfamily of ligand-gated ion channels (also known as pentameric ligand-gated ion channel or pLGICs), which includes nicotinic acetylcholine receptors (nAChRs), 5-HT_3_ receptors (5-HT_3_Rs), and GABA_A_ receptors (GABA_A_Rs) [1,2]. The superfamily members have low sequence homology, but they share a homologous structure in which five subunits pseudo-symmetrically surround a central ion-conducting pore. GlyRs mediate neurotransmission in the brain stem and spinal cord through a glycine-dependent chloride ion influx, causing inhibitory hyperpolarisation in postsynaptic cells [1]. In humans, GlyRs mainly function in pain perception and motor control, with their dysregulation being associated with a range of disorders, including autism and temporal lobe epilepsy [3,4,5]. Mutant GlyRs, however, are most notable for causing human channelopathy hyperekplexia or startle disease.

There are four GlyRα subunit genes (*glra1-4*) and one GlyRβ subunit gene (*glrb*); however, in humans, *glra4* functions as a pseudogene due to the introduction of a premature stop codon. GlyRs consist of α or α and β subunits forming either homomeric α receptors or heteromeric αβ receptors; current evidence suggests an invariant 4α:1β heteromeric stoichiometry for both α_1_β GlyRs and α_2_β GlyRs [6,7]. The presence of β subunits affects pharmacological properties such as picrotoxin efficacy [1], but their major role may be in the binding of intracellular proteins, which control receptor clustering and localisation. The best-studied of these is gephyrin, a 93kDa scaffolding protein which connects GlyRs to the cytoskeleton [8,9]. Gephyrin binds to GlyR β subunits; molecular details of this have been revealed from a crystal structure of a 15-amino-acid GlyR β subunit peptide bound to the C-terminal domain of gephyrin (GephE) [10].

The GlyR, like other pLGICs, has three major domains: the extracellular (ECD), transmembrane (TM), and intracellular (ICD) domain. Many structural studies, however, start by removing the ICD, which is poorly conserved (Figure 1) and flexible. The receptor does function in heterologous systems without this domain, but in vivo it is important as it can modulate a range of characteristics including channel conductance, desensitisation, agonist efficacy, and receptor clustering [8,11,12]. It is also the site of interaction of a range of intracellular proteins, some of which affect assembly and function, while others may regulate downstream signalling pathways. An understanding of the ICD’s structure and function is therefore pivotal in understanding the physiological roles and regulation of GlyRs. This does, however, present a problem, as determining the structure and interactions of large flexible regions of proteins is still very challenging. There has to date only been one structural study on the ICD of a pLGIC—the homomeric α7 nAChR—using nuclear magnetic resonance (NMR) and electron spin resonance experiments combined with Rosetta computations. There are still some problems with these methods of structure determination, in particular the receptor is too large for solution NMR, and thus the study required the use of a suitable smaller construct, which may not fully recapitulate the organisation of the protein in the complete receptor. Nevertheless, these data do provide an experimentally based structure than can be used as a template in protein prediction programs. Some of these programs are highly accurate, much more so than when the acetylcholine-binding protein (AChBP, a pLGIC ECD analogue whose structure was published many years before the structures of any complete receptors were known) was used to model pLGIC ECDs, and even then, relatively imperfect programs provided many useful insights.

Given the importance of the ICD, we wished to explore the structure, stability, flexibility, and interactions of this domain in the GlyR, a pLGIC related to the α7nAChR but which has very different physiological roles. This protein has the advantage of having a relatively well-studied intracellular binding protein—gephyrin—whose C-terminal domain structure, when bound to a GlyR peptide, has been published [10]. However, that study could not determine what affect binding has on protein structure, and indeed, the only method currently available to obtain such information is molecular dynamics experiments. Therefore, here we describe how we used homology-based modelling and molecular dynamics (MD) to create and explore five models of the glycine receptor: one homomeric receptor (5α_1_), two heteromeric receptors (4α_1_:1β and 4α_2_:1β), and two GephE-bound receptors (4α_1_:1β-GephE and 4α_2_:1β-GephE). Using these models, we compared the structure and characteristics of the previously undefined GlyR ICD.

## 2. Materials and Methods

### 2.1. Model Generation

We used ProtCHOIR [13] and MODELLER [14] to generate the GlyR models using the crystal structure of the α3 homomeric human glycine receptor [15] as a template for the α subunits (PDB ID: 5TIN). Signal peptides were removed from structural templates and alignments were performed with MAFFT v 7.515 [16]. As there are not yet any human heteromeric GlyR structures or any GlyR ICDs, three templates were used for each heteromeric model: the Cryo-EM structure of the ECD and TMD (PDB IDs: 7MLY and 5BKF), a ProtCHOIR-generated homomeric α_1_ or α_2_ receptor, and an AlphaFold [17]-predicted GlyR β subunit. All models were in the closed state. The gephyrin-bound models 4α_1_:1β-GephE and 4α_2_:1β-GephE were generated using the GlyR models and the C-terminal GephE domain bound to a GlyR β subunit peptide (PDB ID: 4PD1) using MODELLER and Z-DOCK—a Fast Fourier Transform algorithm which uses electrostatics, desolvation, and shape complementarity to dock rigid bodies [18]. The successful docked models were selected after a qualitative assessment of their structural alignment with 4PD1.

### 2.2. Molecular Dynamics

Each of the five models was inserted into a heterogeneous neuronal plasma membrane (Table 1) using the CHARMM-GUI membrane builder [19,20]. A box of 250 Å × 250 Å was established with periodic boundary conditions. The system was solvated with 150 mM NaCl at a net charge of zero using the TIP3P water model [21]. The simulations were run in GROMACS [22] using the University of Cambridge High Performance Computing Resources. Long-range electrostatic interactions were calculated using the Particle-mesh Ewald method with the Coulomb and van der Waals interaction cut-offs set to 12 Å [23]. The LINCS algorithm was used to constrain molecular bonds. All systems were run using the CHARMM36m force field. Following steepest descent minimisation, all systems were subjected to six series of a 125 picosecond NPT equilibration ensemble with temperature coupling with velocity rescaling and pressure coupling using the Parrinello–Rahman method [24,25,26] for all simulations. Simulation time steps were set at 2 fs, and, as RMSDs plateaued within 10 ns for all models, total simulation runtimes were 50 ns (receptor models) or 25 ns (docked receptor models); simulations were run 3–5 times for each model.

### 2.3. Model Analysis

Visual analysis of GROMACS trajectories was performed using VMD 1.9.4 [27], and snapshots were prepared using PyMOL (The PyMOL Molecular Graphics System, Version 1.2r3pre, Schrödinger, LLC, New York, NY, USA). Snapshots of the ICD were taken from trajectories for each model after equilibration. Stabilising interactions were calculated with RING 3.0, a residue interaction network representation of residue contacts [28]. The default RING 3.0 settings were used with distance thresholds for interactions set at 3.5 Å for hydrogen bonding, 4 Å for ionic, 5 Å for π-cation, 6.5 Å for π-π stacking, 2.5 Å for disulphide, and 0.5 Å for van der Waals. RMSD calculations (Equation (1)) were performed with the gmx rms module using the initial structure as a reference.
(1)$$RMSD\left(t\right)=\sqrt{\frac{1}{N}\;\sum_{i=1}^{N}(r_{i}\left(t\right)-r_{i}^{ref}{)}^{2}}$$
where N is the number of atoms in the model, t is the simulation timepoint, r_i_ is the set of coordinates of an atom, and r_i_^ref^ is the set of coordinates of the same atom in the reference structure.

RMSF (ρ) calculations (Equation (2)) were performed using the gmx rmsf module on Cα atoms in the models. Average coordinates were calculated when the trajectory equilibrated as determined by RMSD values (10 ns—end for all trajectories).
(2)$$\rho_{i}=\sqrt{\langle (r_{i}-\langle r_{i}\rangle {)}^{2}\rangle }$$
where ρ_i_ is RMSF for an atom, r_i_ is the set of coordinates of that atom, and 〈r_i_〉 is the average set of coordinates of that atom.

Polar contacts between GephE and the GlyR β subunit in the docked models were determined by distance. Data were plotted using RStudio v1.3.1903 (RStudio Team (2020). RStudio: Integrated Development for R. RStudio, PBC, Boston, MA, USA URL http://www.rstudio.com/ accessed on 10 March 2023).

### 2.4. Statistics

RMSF data were transformed using a one-parameter Box–Cox power transformation [29] due to a lack of normality violating an assumption of ANOVA.

The transformation involves forming n pairs of sorted data and normalised data (Equation (3)). The value λ, which gives the greatest correlation between the pairs, was then used for the transformation. Data were then transformed using Equation (4).
(3)$$\left(\Phi^{-1}\left(\frac{i-0.5}{n}\right),x_{\left(i\right)}\right),\;for\;i=1,2,\ldots ,n$$
where φ^−1^ is the inverse normal cumulative density function, and x_(i)_ is the ith sorted value.
(4)$$x_{i}^{\lambda }=\left\{\begin{array}{cc} \frac{x_{i}^{\lambda }-1}{\lambda } & if\;\lambda \;\neq 0,\\ ln(x_{i}) & if\;\lambda =0 \end{array}\right.$$

Standard one-way ANOVA and Tukey Honestly Significant Difference tests were performed on transformed RMSF data.

Other data were compared using Student’s test; *p* < 0.05 was taken as significantly different.

## 3. Results

### 3.1. GlyR Models

The GlyR models (Figure 2) show the ECDs and TMDs are similarly structured, while the ICD varies considerably; this is consistent with minimal flexibility in the ECD and TMD but a highly dynamic ICD. Nevertheless, the ICDs of all subunits in each model contained two short helices at the N and C termini, which we call hN and hC: AVNFSR (α_1_ and α_2_, hN), KLFIQRAKKIDK (α_1,_ hC), KFVRAKRIDT (α_2_, hC), VVQVMLNN (β, hN), and PVIPTAAKRIDL (β, hC).

### 3.2. ICD Structure following MD Simulation

Visualisation of the MD trajectories identified a compaction of the ICDs over time (Figure 3). ICD compaction was most pronounced in the 5α_1_ model (its maximum width was reduced from 115 ± 8 Å to 69 ± 4 Å, mean ± SEM, *n* = 3), perhaps due to its initial pseudosymmetry and the absence of the larger β subunit ICD. The 4α_2_:1β ICD underwent less compaction, but it also had the smallest diameter in its 0 ns structure. Pore diameter was not affected in any of the models and was relatively constant at ~30 Å.

The MD data also revealed that various secondary structures formed transiently, e.g., in the β subunit of the 4α_1_:1β GlyR, a coil observed at ^421^SIV^423^ in the initial structure was lost over the trajectory, while in the β subunit of 4α_2_:1β GlyR, a small helix was formed after 50 ns at ^406^KKVCT^410^, as was a tripeptide β sheet at ^411^SKS^413^–^423^VIS^421^.

### 3.3. ICD Interactions

Interactions that could stabilise the ICD were explored during the MD simulation for each model (Figure 4). These were predominantly van der Waals interactions and hydrogen bonds. For the unbound models, these increased during the simulation, but there was no increase in the docked models 4α_1_:1β-GephE and 4α_2_:1β.-GephE Given the paucity of data on pLGIC ICDs, it is not yet clear which interactions are likely to predominate in stabilising intracellular loops. These data do, however, suggest that the presence of GephE could stabilise the whole ICD.

### 3.4. ICD β Subunit Flexibility

The RMSF data showed that the α subunit influenced the flexibility of the β subunit ICD: The β ICD in 4α_2_:1β GlyR had a significantly larger RMSF, indicating greater flexibility in the β ICD here compared to the 4α_1_:1β GlyR (Figure 5).

### 3.5. Effects of GephE Binding on ICD Structure and Stability

GephE binding did not change the ICD pore diameters, which were 29.5 ± 0.6 Å versus 29.9 ± 0.5 Å for 4α1:1β and 4α1:1β-GephE respectively, and 29.1 ± 0.3Å versus 28.6 ± 0.5 Å for 4α2:1β and 4α2:1β-GephE (data = mean ± SEM, n = 3). GephE binding did, however, reduce the maximum ICD width in the 4α2:1β GlyR, which decreased from 120.9 ± 5.1 Å to 86.7 ± 4.1 Å (significantly different *p* < 0.01). Width values for the 4α1:1β GlyR were not significantly different (81.2 ± 21 Å and 78.7 ± 22 Å with and without GephE, respectively), but observation of the bound models did reveal compaction (Figure 6); the diameter measurements did not reflect this as one loop of one α_1_ subunit was extended despite the compaction of the other four subunits. Thus, the changes in structure were quite different for the two heteromeric receptors, illustrating that subtle differences in the binding of the same molecule to the same subunit in similar receptors could have very different outcomes.

A closer examination of the site of interaction between GlyR and GephE revealed that the 4α_2_:1β-GephE binding site has more possible interactions than the 4α_1_:1β-GephE binding site (Figure 7), thus hinting at the fact that there could be differences in affinity for gephyrin in the different heteromers.

## 4. Discussion

This study provides a predictive insight into the previously unknown structure and characteristics of the GlyR ICD and also shows that the ICD has the potential to undergo considerable conformational changes upon the binding of physiologically relevant intracellular proteins such as gephyrin. Structural studies to date have not been able to generate high-resolution structural details of this domain, but using molecular details from a related protein and the power of modelling and molecular dynamics, we have produced what is likely to be a reasonably, if not fully, accurate structure of the GlyR ICD.

The details revealed by our GlyR models explain why this region has proved incalcitrant to structural experiments: much of it comprises a highly flexible loop, and this, combined with its considerable size and variability, is not conducive to obtaining good experimental data from classic structural techniques such as X-ray crystallography, cryo-electron microscopy and NMR. Computer-based studies are the only practical route to obtaining molecular information, and in using these, we observed a range of useful structural details. These include the fact that the ICDs of both α and β subunits form helices at their N and C ends, and the flexible loop between these helices can support transient secondary structures. We show that there is greater flexibility of the β subunit in the 4α_2_:1β GlyR compared to the 4α_1_:1β GlyR, which suggests the possibility of different binding partners of the β subunit depending on the other receptor subunits. In addition, our data predict the effect and molecular details of gephyrin binding to the GlyR ICD. In our GephE-bound GlyRs, the ICDs are more compact, and the number of potential residue interactions between GephE and the β subunit differ in the 4α_2_:1β GlyR compared to the 4α_1_:1β GlyR. Thus, not only do our data suggest that the β subunit could have different binding partners depending on the α subunits but also that the same binding partner could generate different conformational changes in different heteromers. The resulting different functional effects could help explain the reasons behind the differential expression of GlyR subunits in different tissues and over developmental timescales [30,31].

Our ICD models are broadly similar to the structure of the ICD of the α_7_ nAChR ICD, which has recently been investigated [32]. However, when compared to the results presented here, the α7 nAChR ICD has additional secondary structures: each subunit contains one C-terminal MA helix, one N-terminal MX helix, and three short helices (h1–h3). The h3 helix anchors the MA helix, resulting in a B-shaped architecture. We did find C-terminal and N-terminal helices in the GlyR ICD, but these were much shorter than the MA and MX helices. We did not observe any consistent helices equivalent to h1–h3 in the GlyR ICD, although analysis using the secondary-structure prediction program PSIPRED [33] did predict a number of short helices and strands which largely occurred in positions similar to the transient secondary structures we observed. Thus, our data suggest that GlyR ICDs are more disordered than those of α_7_ nAChR, with fewer fixed secondary structures.

Disorder is likely to be an important feature of Cys-loop receptor ICDs. Disorder is known to be prevalent across eukaryotic proteins, particularly signalling proteins [34,35], and it may provide an evolutionary advantage by allowing binding to a wide variety of partners [36]. To date, four intracellular binding partners of GlyR ICDs have been identified: gephyrin, neurobeachin, vacuolar protein sorting 35, and syndapin I [37,38,39]. All of these proteins bind to the β subunit in the ICD, with only one—syndapin I—also binding to the α subunit. Binding to disordered regions often induces a conformational change [40], and we propose that this would be the case when these partners bind to the GlyR ICD. This hypothesis is supported by our models, which show ICD compaction when GephE is bound. A conformational change in the ICD could also affect other regions of the protein, causing changes to the function and/or interactions of the receptor.

All the results presented here were generated using a computational approach involving ab initio model generation and MD simulations. There are still a number of well-recorded potential errors in these and related software programs, which have particular difficulty in generating multimeric proteins and loop regions [41,42]. Nevertheless, in silico approaches have proved useful and are continually improving; not only have they led to increasingly accurate predictions, but they are often the only practical route to obtaining data. Here, an in silico method was necessary due to the size and intrinsic disorder of GlyR ICDs, which have largely prevented experimentally solved structures from being useful; some previous experimental attempts have, for example, resulted in low-resolution “blob-ology” [6]. Our models and simulations are therefore useful in creating hypotheses which can be further tested, and they provide a starting point for future research and development.

## 5. Conclusions

In conclusion, this study reveals likely features of the previously unknown structure, stability, and flexibility of the GlyR ICD. As no structures of this region in the GlyR are currently available, we used a bioinformatic approach to generate plausible models of the ICDs of three different GlyRs: one homomeric receptor (5α_1_) and two heteromeric receptors (4α_1_:1β and 4α_2_:1β). These were then subjected to MD simulations to improve accuracy. The outputs revealed more compact structures than our initial models suggested, with an increased number of intracellular interactions. Our data also indicated that the type of α subunit could influence the flexibility of the β subunit, with the β subunit ICD in 4α2β1 receptors likely having greater flexibility compared to the 4α1β1 GlyR. This structural diversity of GlyR ICDs may be related to the functional diversity of different GlyR isoforms.

We also generated two models of GlyRs bound to gephyrin. Gephyrin is an intracellular protein and has been well documented as a GlyR-binding protein which controls receptor localisation and clustering. It has three important domains: a C domain, an E domain, and a G domain. The C domain is important for localisation, as it binds to microtubules, while the E and G domains are important for clustering: the E domain can dimerise and the G domain can trimerise. These gephyrin oligomerisations via the E and/or G domains cause GlyR clustering through the formation of sub-membranous, hexagonal lattices. Gephyrin binds to GlyR β subunits; molecular details of this have been revealed from a crystal structure of a 15-amino-acid β subunit peptide from gephyrin bound to GlyRs via its E domain [10]. We used these data to help generate two models of the GlyR bound to Geph.: 4α_1_:1β-GephE and 4α_2_:1β-GephE. MD simulations of these models revealed no increase in total ICD binding interactions, indicating that GephE had sufficiently stabilised the original structures. However, a visual inspection revealed that the bound GlyR was more compact in at least four of its five subunits. There was also evidence that the different α subunits influence the binding differently: the 4α_2_:1β GlyR-GephE binding site has more potential interactions than the 4α_1_:1β GlyR-GephE binding site, indicating possible different affinities for gephyrin in the different heteromers. Differences in receptor compaction and binding affinity could be transduced to other regions of the receptor to generate different functional effects.

Thus, overall, our study has enhanced our knowledge of the structure and interactions of the GlyR and revealed a range of features that could explain its different functions, especially those that involve binding to its best-studied interaction partner, gephyrin.

## Figures and Tables

**Figure 1 biomolecules-13-01757-f001:**
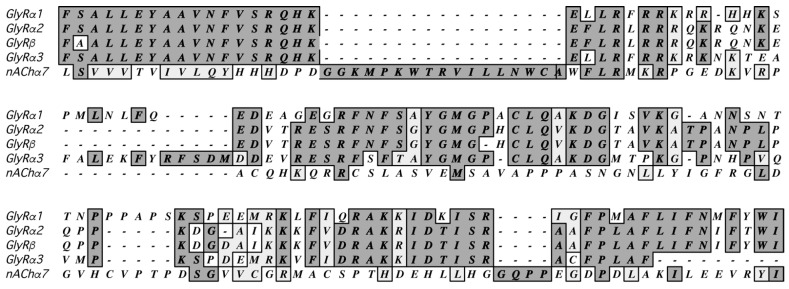
A Clustal Omega sequence alignment of the ICD reveals low sequence similarity between human GlyR subunits and also between other human pLGIC subunits, such as the α_7_ nAChR subunit. Residue identity and lone residues are shown in dark grey, and residues with similar chemical properties are shown in light grey. Residue numbers are GlyRα1: 299–444; α2: 306–439; β: 329–438; α3: 328–430; nAChRα7: 318–469.

**Figure 2 biomolecules-13-01757-f002:**
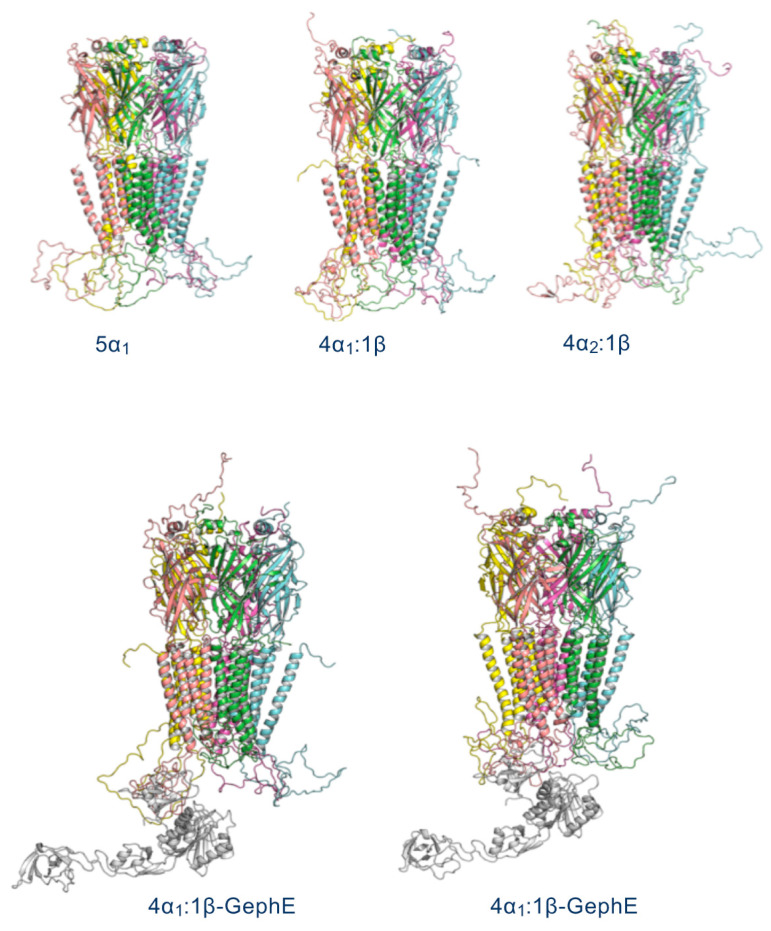
GlyR models. Colours distinguish subunits, with light pink (leftmost subunit) representing β in heteromeric models.

**Figure 3 biomolecules-13-01757-f003:**
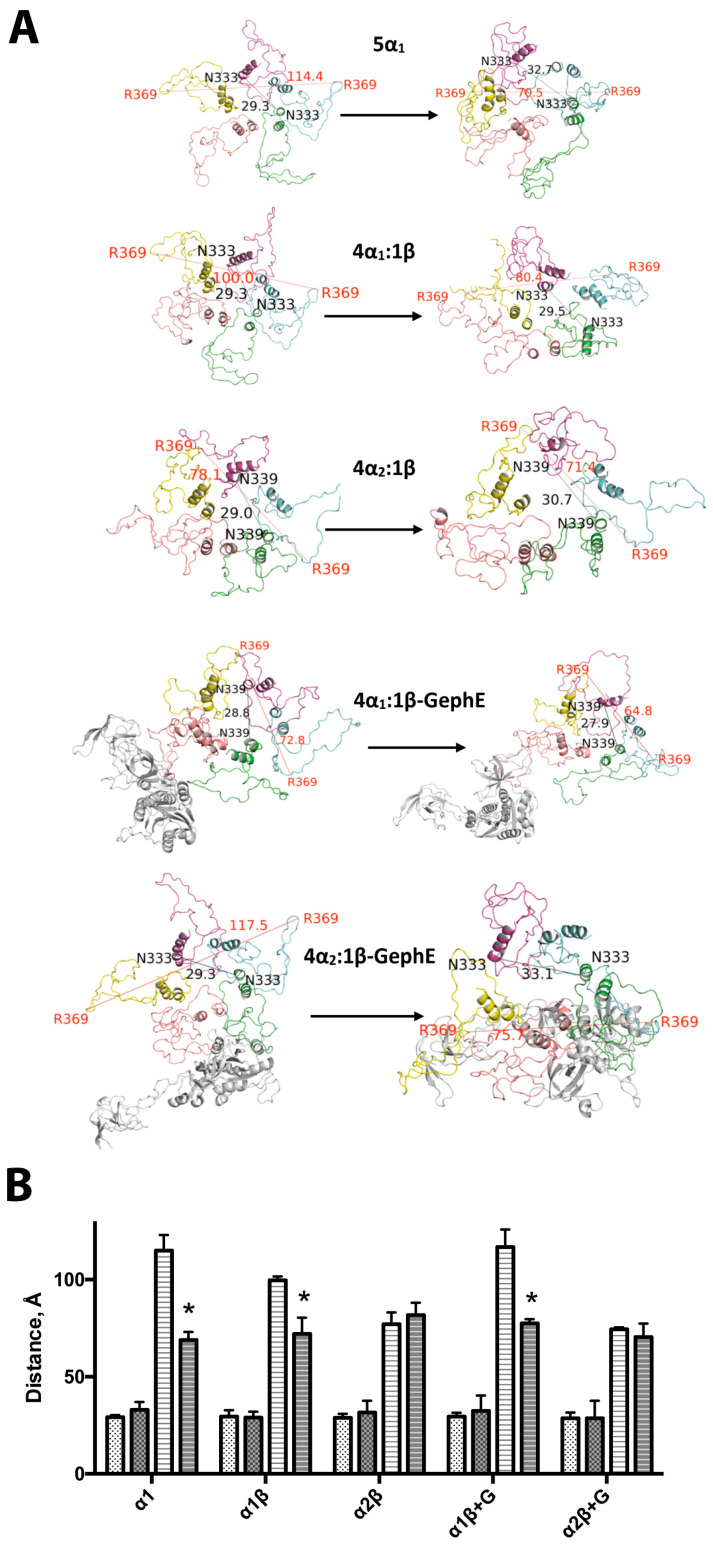
Changes in ICD structures after MD simulations. (**A**) Example structures from before (left-hand side) and after (right-hand side) the simulation, viewed from the bottom of the receptor. Typical of 3–5 production runs. The maximal ICD width (measured between Cα in Å) is labelled in red, and the pore diameter (between Cα in Å) in black. (**B**) Pore (spots) and diameter (lines) distances before (white) and after (grey) the MD simulations. * = significantly different, Student’s *t* test, *p* < 0.05. Data = mean ± SEM, *n* = 3.

**Figure 4 biomolecules-13-01757-f004:**
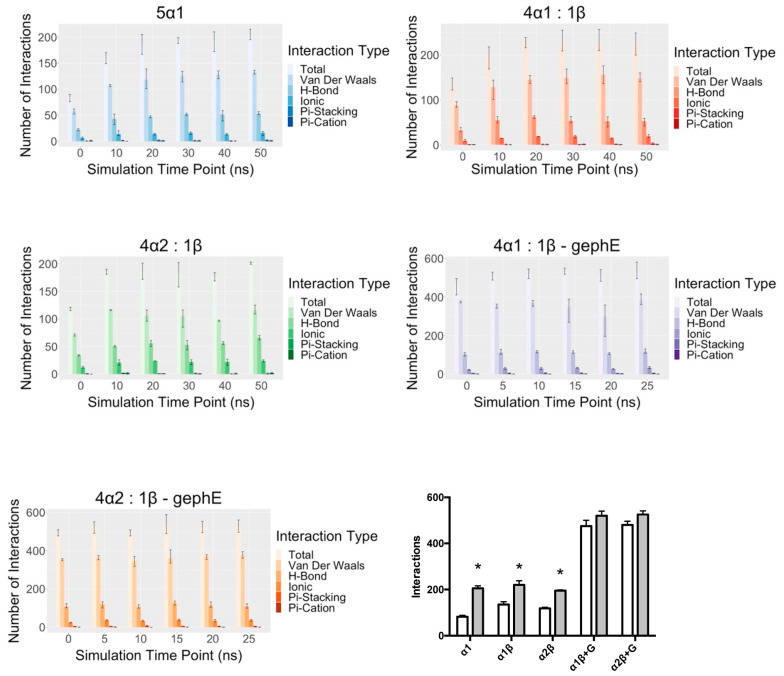
Changes in ICD interactions during the MD simulation for each model. A range of different interactions were identified and are shown in different shades of colour. For the unbound GlyR models (5α_1,_ 4α_2_:1β, and 4α_2_:1β), the number of interactions increased during the simulation, but there was no significant increase in interactions in the bound models (4α_1_:1β-gephE and 4α_2_:1β-gephE). This can be seen in the plot at the bottom right-hand side, which shows the total number of interactions at the start (white) and end (grey) of the simulations. * = significantly different to start, Student’s t test, *p* < 0.05. Data = mean ± SEM, n = 3.

**Figure 5 biomolecules-13-01757-f005:**
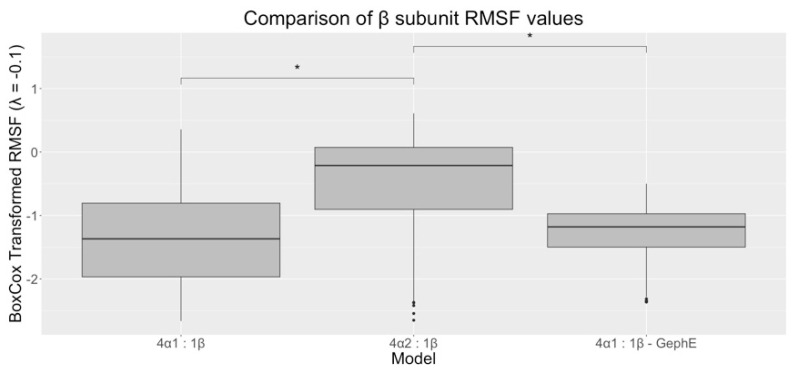
Comparison of Box–Cox transformed β subunit ICD RMSF values. The 4α_2_:1β GlyR model had a larger RMSF value than 4α_1_:1β and 4α_1_:1β-GephE GlyRs. * = significantly different at *p* < 0.0001, n = 3.

**Figure 6 biomolecules-13-01757-f006:**
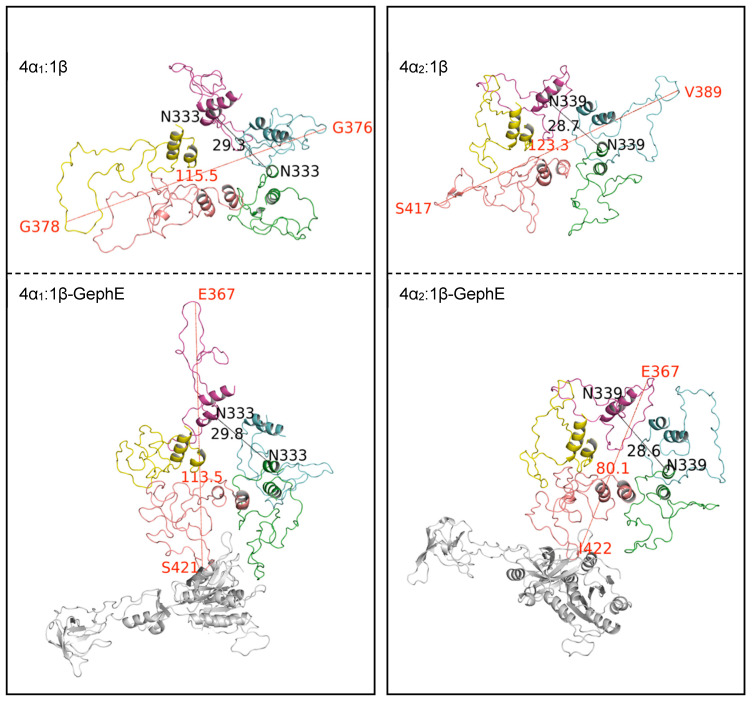
The effect of GephE binding on ICD structure. Images of ICDs without (upper panel) and with (lower panel) bound GephE. Typical of 3 production runs. Maximal ICD width (red) and pore diameter (black) in Å are shown.

**Figure 7 biomolecules-13-01757-f007:**
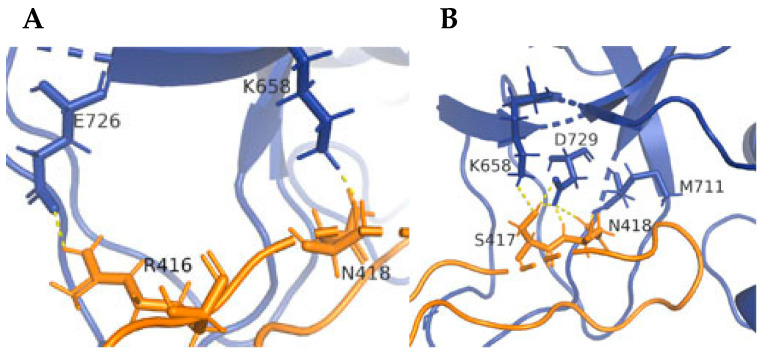
Residues involved in GephE binding may differ for 4α_1_:1β-GephE (**A**) and 4α_2_:1β-GephE (**B**). Potential interactions between gephyrin (blue) and the α subunit (orange) following a typical MD production run are shown as dashed yellow lines. More potential interactions are observed in the 4α_2_:1β-GephE model.

**Table 1 biomolecules-13-01757-t001:** Lipid composition of the neuronal plasma membrane.

Lipid	% Upper Leaf	% Lower Leaf
Cholesterol	44	44
DPPC	13	6
POPC	21	10
DOPC	6	3
POPE	16	21
POPS	0	10
POPI	0	5
POPA	0	1
Total	100	100

## Data Availability

Data are available on request from the authors.

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
