# Peer review of "Modelling and Molecular Dynamics Predict the Structure and Interactions of the Glycine Receptor Intracellular Domain"

_biomolecules, 2023, doi:10.3390/biom13121757_

Round 1

Reviewer 1 Report

Comments and Suggestions for Authors

The work is devoted to the study of the structure and its influence on the functions of the glycine receptor. Inhibitory glycine receptors are anion-selective ligand-gated ion channels (LGICs), a member of the eukaryotic Cys-loop family. Disturbances in the functioning of the glycine receptor are associated with a number of socially significant diseases. Unfortunately, the complete three-dimensional structure of glycine receptors has not yet been determined. This article is intended to supplement knowledge about the structure and functioning of the receptor, namely its intracellular domain. The article will be of undoubted interest to the readers of the journal. This paper is recommended for publication with minor revision.

Some remarks:

1. It should be clarified which living organism the primary sequences in Figure 1 refer to. Please also add the amino acid residue numbers of the beginning and end of the sequences.

2. The authors examine the alpha-3 subunit, but its sequence is not shown in Figure1. It would be clearer if Figure 1 were expanded to add alpha-3 to the alignment, indicating a living organism and residue numbers.

3. Lines 86-89. Models of subunits of different living organisms are used as templates (Homo sapiens (5TIN), Rattus norvegicus, Sus scrofa (7MLY), Homo sapiens, Aequorea victoria (5BKF)). How justified is this approach and is it legitimate to use such models? What living organism are the authors trying to build a model of glycine receptors? This section should be supplemented with this information.

4. Line 90-93. The gephyrin bound models

What parameters were used and how many gephyrin-receptor models were selected for the simulation? The description does not allow the experiment to be repeated.

5. Line 95. Each of the five models

Please be specific about which five models are being studied. Were the receptors studied in an open or closed state? How was the state of the receptor controlled in the simulation?

6. Lines 104-106. temperature coupling using velocity rescaling and pressure coupling using the Parrinello-Rahman method

Were these parameters the same for the MD simulations?

7. Lines 106-107. Simulation time steps were set at 2fs for a total simulation  runtime of either 50ns (receptor models) or 25ns (docked receptor models).

The lengths of the trajectories do not seem large. How was it determined that they were sufficient to obtain accurate results?

8. Lines 107-108. simulations were run 3-5 times for each model

It is not clear what exactly was simulated 3-5 times. What were the differences?

9. Lines 179-181. MD simulations reduced the number of hydrogen bonds and increased the number of ionic interactions in comparison to models before MD.

The data in Figure 4 is inconsistent with these claims. The number of hydrogen bonds increases. The number of ionic interactions is difficult to estimate in figures of this scale. However, it seems that their increase is insignificant and is within the limits of measurement error.

10. Line 185. Data = mean +- SEM, n = 3.

The expression contains an incomprehensible abbreviation and requires decoding.

11. Lines 209-212. the 4α2:1β-GephE binding site has more possible interactions than the 4α1:1β-GephE

The data obtained is interesting. Is it possible to estimate the differences in binding free energy of gephyrin to complexes?

12. Line 218. Discussion.

The data obtained in the article are of undoubted interest. However, in the discussion, again, there is no indication whether they refer to the receptor of a specific living organism or of all organisms? How do the obtained data compare with known experimental data? The abstract mentions the effect of receptor clustering through gephyrin. However, the article and discussion do not contain any data on this effect.

Author Response

The work is devoted to the study of the structure and its influence on the functions of the glycine receptor. Inhibitory glycine receptors are anion-selective ligand-gated ion channels (LGICs), a member of the eukaryotic Cys-loop family. Disturbances in the functioning of the glycine receptor are associated with a number of socially significant diseases. Unfortunately, the complete three-dimensional structure of glycine receptors has not yet been determined. This article is intended to supplement knowledge about the structure and functioning of the receptor, namely its intracellular domain. The article will be of undoubted interest to the readers of the journal. This paper is recommended for publication with minor revision.

Some remarks:

  1. It should be clarified which living organism the primary sequences in Figure 1 refer to. Please also add the amino acid residue numbers of the beginning and end of the sequences.

We have now revised this figure and added these details to its legend.

  1. The authors examine the alpha-3 subunit, but its sequence is not shown in Figure1. It would be clearer if Figure 1 were expanded to add alpha-3 to the alignment, indicating a living organism and residue numbers.

This has been added in the revised version

  1. Lines 86-89. Models of subunits of different living organisms are used as templates (Homo sapiens (5TIN), Rattus norvegicus, Sus scrofa (7MLY), Homo sapiens, Aequorea victoria (5BKF)). How justified is this approach and is it legitimate to use such models? What living organism are the authors trying to build a model of glycine receptors? This section should be supplemented with this information.

Our aim was to obtain models as close as possible to human GlyRs but as we do not yet have high resolution human heteromeric GlyR structures or any GlyR ICDs, we used other templates/structures for better model accuracy which is a common approach.  We have now clarified this in the revised manuscript.

  1. Line 90-93. The gephyrin bound models

What parameters were used and how many gephyrin-receptor models were selected for the simulation? The description does not allow the experiment to be repeated.

These details have now been added 

  1. Line 95. Each of the five models

Please be specific about which five models are being studied. Were the receptors studied in an open or closed state? How was the state of the receptor controlled in the simulation?

All our 5 models were studied and all were in the closed state at both the start and end of the simulation without any control being applied. We have clarified this in the revised manuscript. 

  1. Lines 104-106. temperature coupling using velocity rescaling and pressure coupling using the Parrinello-Rahman method

Were these parameters the same for the MD simulations?

Yes. We have clarified this in the revised manuscript. 

  1. Lines 106-107. Simulation time steps were set at 2fs for a total simulation runtime of either 50ns (receptor models) or 25ns (docked receptor models).

The lengths of the trajectories do not seem large. How was it determined that they were sufficient to obtain accurate results?

We judged accuracy by ensuring that the RMSD had reached equilibrium, which occurred within 10ns for all of the models ( now stated in the revised manuscript) .  We agree that coarse-grained molecular dynamics to investigate the effects over longer timescales would be useful for future studies such as to determine potential structural changes when the receptors are activated; however, this study was primarily focused on residue-residue contacts involved in ICD conformations in the closed state.

  1. Lines 107-108. simulations were run 3-5 times for each model

It is not clear what exactly was simulated 3-5 times. What were the differences?

There were no differences in the input and indeed the outputs from repeated simulations were very similar;  nevertheless we wished to repeat each several times to ensure representative data.

  1. Lines 179-181. MD simulations reduced the number of hydrogen bonds and increased the number of ionic interactions in comparison to models before MD.

The data in Figure 4 is inconsistent with these claims. The number of hydrogen bonds increases. The number of ionic interactions is difficult to estimate in figures of this scale. However, it seems that their increase is insignificant and is within the limits of measurement error.

We have corrected this sentence and now explain that the number of interactions increases over time for the for a1, a1b and a2b models but not for the gephyrin bound models. 

  1. Line 185. Data = mean +- SEM, n = 3.

The expression contains an incomprehensible abbreviation and requires decoding.

SEM is a standard mathematical abbreviation  ( standard error of the mean) although the minus sign seems to have slipped here.

  1. Lines 209-212. the 4α2:1β-GephE binding site has more possible interactions than the 4α1:1β-GephE

The data obtained is interesting. Is it possible to estimate the differences in binding free energy of gephyrin to complexes?

This is an interesting idea and something we would like to do in the future but it is beyond the scope  of this study.

  1. Line 218. Discussion.

The data obtained in the article are of undoubted interest. However, in the discussion, again, there is no indication whether they refer to the receptor of a specific living organism or of all organisms? How do the obtained data compare with known experimental data? The abstract mentions the effect of receptor clustering through gephyrin. However, the article and discussion do not contain any data on this effect.

We consider our receptors best represent human GlyR, although are likely to be similar in other organisms, and this information has now been added to the manuscript. However it is difficult to compare our simulations with experimental data because  - as far as we are aware – there have been no studies on the ICD of the GlyR. Regarding the clustering, we did not generate any data that would address this, but it would be an interesting aspect to follow up in future studies with longer trajectories.

Reviewer 2 Report

Comments and Suggestions for Authors

The Authors study two hetero-pentameric versions of the human glycine receptor (GlyR) family of proteins using molecular dynamics. The key question asked was the role of the intracellular domain in the interactions between the receptor and gephyrin protein. The results show the difference in interactions between receptor and gephyrin for the different members of the GlyR family and conclude that this might be indicative of finetuning of GlyRs signaling.

The text is well written and the conclusion are mostly sound. The only  scientific question is regarding the usage of the CHARMM36 force field. Why did the Authors use this instead of the newer CHARMM36m (Nature Methods volume 14, pages 71–73 (2017), already cited in the paper) which is more suited to the studied disordered protein interactions? This decision led to an overrepresentation of ionic interactions, a problem solved in the new force field. 

The graphical presentation still needs work. 

Issues:

Figure1. Row 3 contains GlyRα instead of β. What are the dark-grey residues which do not align with any other?

Figure 2. Are these the starting models? It is not clear for me

Figure 3. The annotations on the protein chains are hard to see, especially if overlapped with protein. The residues are labeled are not defined as belonging to one or the other chain. Please at least indicate chains in the residue names. GephE has the p bottom cut off . Please correct.

In general the quality of the images should be improved.

Author Response

The Authors study two hetero-pentameric versions of the human glycine receptor (GlyR) family of proteins using molecular dynamics. The key question asked was the role of the intracellular domain in the interactions between the receptor and gephyrin protein. The results show the difference in interactions between receptor and gephyrin for the different members of the GlyR family and conclude that this might be indicative of finetuning of GlyRs signaling.

The text is well written and the conclusion are mostly sound. The only  scientific question is regarding the usage of the CHARMM36 force field. Why did the Authors use this instead of the newer CHARMM36m (Nature Methods volume 14, pages 71–73 (2017), already cited in the paper) which is more suited to the studied disordered protein interactions? This decision led to an overrepresentation of ionic interactions, a problem solved in the new force field. 

We did use this version but unfortunately omitted the m in the methods section, even though we added the correct reference there. This has now been corrected.

The graphical presentation still needs work. 

We have redone a number of the images

Issues:

Figure1. Row 3 contains GlyRα instead of β. What are the dark-grey residues which do not align with any other?

This has now been corrected/clarified in the revised manuscript.

Figure 2. Are these the starting models? It is not clear for me

Yes. This has now been clarified in the revised manuscript

Figure 3. The annotations on the protein chains are hard to see, especially if overlapped with protein. The residues are labeled are not defined as belonging to one or the other chain. Please at least indicate chains in the residue names. GephE has the p bottom cut off . Please correct.

In general the quality of the images should be improved.

We have now redone a number of the images including figure 3

Reviewer 3 Report

Comments and Suggestions for Authors

The manuscript by Thompson et. al. describes the generation of models for the receptor GlyR in different combinations of alpha/beta subunits, as well as complexes of the receptor with the intracellular protein Geph bound to the ICD. Using those models, authors analyzed the conformational dynamics of the ICD in MD simulations. The manuscript provides new relevant information but requires multiple corrections before it possible publication.

Specific comments:

1.     Figure 1. Residues unique to one receptor are shown in dark gray. That is a mistake since those residues cannot be compared.

2.     Simulation times (50 ns and 25 ns) seem short, even for large complexes within membrane. Please extend at least one replica of key experiments.

3.     Although the main goal of the work is analyzing the ICD, it seems (Fig. 3) that the TM helices are changing their axe. Did authors study the regions adjacent to the ICD?

4.     In section 3.2 authors provide the maximal ICD width to support its compaction. Since the data comes from dynamic studies, I suggest providing a mean with dispersion and kinetics graphs. It is unclear what is “typical” (Fig. 3) if you do not provide data for the replicas.

5.     The above comment also applies to the number of interactions described in 3.3 and data presented in Fig. 6.

6.     From a statistical point of view, samples with 170 and 200 units (3.3 and Fig. 4) could be part of the same population. Did you perform statistical analysis? Looking at the data of Fig. 4, I disagree that “the number of interactions in the docked models also increased during the simulations”, especially for 4alpha2:1beta-Geph.

7.   Is a type of interaction more important in stabilizing intracellular loops? Please discuss.

8.     Please increase resolution of Fig. 7.

9.     Generate alanine mutants for residues proposed to be key in mediating GlyR-Geph interactions and run the corresponding MD simulaiton.

Author Response

The manuscript by Thompson et. al. describes the generation of models for the receptor GlyR in different combinations of alpha/beta subunits, as well as complexes of the receptor with the intracellular protein Geph bound to the ICD. Using those models, authors analyzed the conformational dynamics of the ICD in MD simulations. The manuscript provides new relevant information but requires multiple corrections before it possible publication.

Specific comments:

  1. Figure 1. Residues unique to one receptor are shown in dark gray. That is a mistake since those residues cannot be compared.

This has now been corrected/clarified in the revised manuscript.

  1. Simulation times (50 ns and 25 ns) seem short, even for large complexes within membrane. Please extend at least one replica of key experiments.

We judged accuracy by ensuring that the RMSD had reached equilibrium, which occurred within 10ns for all of the models ( now added in the revised manuscript).  We agree that coarse-grained molecular dynamics to investigate the effects over longer timescales would be useful for future studies such as to determine potential structural changes when the receptors are activated; however, this study was primarily focused on residue-residue contacts involved in ICD conformations in the closed state.

  1. Although the main goal of the work is analyzing the ICD, it seems (Fig. 3) that the TM helices are changing their axe. Did authors study the regions adjacent to the ICD?

This is an interesting observation  and something we hope to do in the future but is beyond the scope of this study.

  1. In section 3.2 authors provide the maximal ICD width to support its compaction. Since the data comes from dynamic studies, I suggest providing a mean with dispersion and kinetics graphs. It is unclear what is “typical” (Fig. 3) if you do not provide data for the replicas.

And

  1. The above comment also applies to the number of interactions described in 3.3 and data presented in Fig. 6.

This is a good idea. These data have now been provided in a new graph and are discussed in  the revised manuscript.

  1. From a statistical point of view, samples with 170 and 200 units (3.3 and Fig. 4) could be part of the same population. Did you perform statistical analysis? Looking at the data of Fig. 4, I disagree that “the number of interactions in the docked models also increased during the simulations”, especially for 4alpha2:1beta-Geph.

We have now done the statistical analysis as sensibly suggested by the reviewer . These data show that the number of interactions increases over time for the for a1, a1b and a2b models but not for the gephyrin bound models, and so we have revised this section. .

  1. Is a type of interaction more important in stabilizing intracellular loops? Please discuss.

We have added some discussion of this in the ICD interactions section.

  1. Please increase resolution of Fig. 7.

Now increased.

  1. Generate alanine mutants for residues proposed to be key in mediating GlyR-Geph interactions and run the corresponding MD simulaiton.

This is a good idea and something we would like to do in the future, but it is beyond the scope of this study.